# Nutritional control of body size through FoxO-Ultraspiracle mediated ecdysone biosynthesis

Takashi Koyama[1]*, Marisa A Rodrigues[1], Alekos Athanasiadis[2], Alexander W Shingleton[3,4], Christen K Mirth[1]*

[1]Development, Evolution and the Environment Laboratory, Instituto Gulbenkian de Ciência, Oeiras, Portugal; [2]Protein-Nucleic Acids Interactions Laboratory, Instituto Gulbenkian de Ciência, Oeiras, Portugal; [3]Department of Biology, Lake Forest College, Lake Forest, United States; [4]Department of Zoology, Michigan State University, East Lansing, United States

**Abstract** Despite their fundamental importance for body size regulation, the mechanisms that stop growth are poorly understood. In *Drosophila melanogaster*, growth ceases in response to a peak of the molting hormone ecdysone that coincides with a nutrition-dependent checkpoint, critical weight. Previous studies indicate that insulin/insulin-like growth factor signaling (IIS)/Target of Rapamycin (TOR) signaling in the prothoracic glands (PGs) regulates ecdysone biosynthesis and critical weight. Here we elucidate a mechanism through which this occurs. We show that Forkhead Box class O (FoxO), a negative regulator of IIS/TOR, directly interacts with Ultraspiracle (Usp), part of the ecdysone receptor. While overexpressing FoxO in the PGs delays ecdysone biosynthesis and critical weight, disrupting FoxO–Usp binding reduces these delays. Further, feeding ecdysone to larvae eliminates the effects of critical weight. Thus, nutrition controls ecdysone biosynthesis partially via FoxO–Usp prior to critical weight, ensuring that growth only stops once larvae have achieved a target nutritional status.

*For correspondence: tkoyama@igc.gulbenkian.pt (TK); christen@igc.gulbenkian.pt (CKM)

**Competing interests:** The authors declare that no competing interests exist.

**Reviewing editor**: Helen McNeill, The Samuel Lunenfeld Research Institute, Canada

## Introduction

Environmental conditions mould the developmental programs of many organisms to produce dramatic differences in body size and shape, in developmental time and in pigmentation patterns (*Beldade et al., 2011*). In insects, environmental cues often mediate their effects by regulating the timing and amount of hormone biosynthesis at specific points in development (*Koyama et al., 2013*). These changes in hormone production have been associated with a wide variety of environmentally induced changes in morphology, including the dramatic reshaping of the body in honeybee castes and seasonal wing pattern polyphenisms in butterflies (*Beldade et al., 2011*; *Koyama et al., 2013*). Understanding the molecular underpinnings through which environmental conditions modify hormone production would provide valuable insight into our understanding of developmental plasticity.

Larvae of the fruit fly, *Drosophila melanogaster*, provide a tractable model to address this question. *Drosophila* larvae regulate their body size and developmental timing in response to nutritional conditions, similar to many other animals (*Shingleton, 2011*). Early in the third (final) larval instar (L3), a small peak of the steroid hormone ecdysone has been proposed to induce a developmental transition known as critical weight (*Mirth and Riddiford, 2007*; *Mirth and Shingleton, 2012*). The critical weight ecdysone peak responds to both environmental cues and internal developmental processes. Environmental cues including nutrition, temperature and oxygen levels affect the timing of the critical weight ecdysone peak (*Caldwell et al., 2005*; *Colombani et al., 2005*; *Mirth et al., 2005*;

**eLife digest** The development of many of an animal's traits and characteristics are sensitive to the conditions of its environment. The conditions experienced early on in life, in particular, can alter how an animal grows and develops. The availability of food during development, for example, affects the final body size of many animals—including the larvae of fruit flies, which must reach a so-called 'critical weight' before they can change into adults.

A hormone called ecdysone controls when a larva turns into an adult insect. This hormone's levels peak when a fruit fly larva has reached its critical weight, and this peak triggers a cascade of events that ultimately will cause the larva to stop growing and to change into an adult fly. A signaling pathway involving insulin is also known to help regulate body size in response to nutrition. But it was unclear how the timing, and size, of the ecdysone peak is altered to match the larva's diet—and how the insulin-related pathway exerts its effect on body size.

Now, Koyama et al. have found a molecular link between this signaling pathway and the ecdysone hormone that can explain how nutrition can regulate the growth of the flies. First, Koyama et al. found that the ecdysone peak is delayed when young fruit fly larvae were starved before they had reached the critical weight. On the other hand, feeding these starved larvae the ecdysone hormone prevented this delay; and many of these larvae developed into underweight adults that were about three-quarters the size of a typical fully-grown adult.

The ecdysone hormone is made by cells within certain glands in the larvae. Koyama et al. also found that a protein called FoxO, which inhibits the insulin-related pathway, is transported out of the nuclei of these gland cells when growing larvae gain weight. But when larvae that had not reached their critical weight were starved, the FoxO protein was kept within the cell nuclei. Koyama et al. found that, in the nucleus, the FoxO protein blocks the production of ecdysone by interacting with a protein that forms part of the ecdysone receptor. This protein complex delays the expression of genes that are involved in making the hormone. When larvae were fed, the FoxO protein began to leave the nucleus, which allowed ecdysone production to resume.

Future studies could now test whether FoxO controls other traits that are affected by the insect's diet (such as lifespan), and if growth-related hormones in other animals are affected by a similar mechanism.

*Callier et al., 2013*; *Ghosh et al., 2013*). In addition, the neuropeptide important for inducing all ecdysone peaks, prothoracicotropic hormone (PTTH), stimulates ecdysone biosynthesis at critical weight (*McBrayer et al., 2007*; *Ou et al., 2011*). The combination of environmental and developmental regulation of this ecdysone peak ensures that developmental timing can be altered with changes in environmental conditions (*Gibbens et al., 2011*; *Mirth and Shingleton, 2012*).

Critical weight itself determines the duration of the growth period, and therefore final body size, in response to environmental conditions including nutrition (*Beadle et al., 1938*; *Nijhout and Williams, 1974b*; *Mirth et al., 2005*; *Shingleton et al., 2005*; *Stieper et al., 2008*). Before larvae reach critical weight, starvation delays the onset of metamorphosis. After critical weight, larvae initiate metamorphosis without any developmental delay even when starved. Feeding ecdysone to larvae with genetically-induced delays in critical weight rescues the timing of the onset of metamorphosis (*Stieper et al., 2008*; *Parker and Shingleton, 2011*).

Nutrition regulates size in organisms ranging from flies to humans via the insulin/insulin-like growth factor signaling (IIS)/Target of Rapamycin (TOR) signaling pathway (*Grewal, 2009*). The IIS/TOR pathway controls body size by regulating growth rate, and also by regulating the timing of critical weight to determine the duration of the growth period (*Nijhout, 2003*). At critical weight, the IIS/TOR pathway acts directly on the glands that synthesize ecdysone, the prothoracic glands (PGs), to alter the timing of the expression of several cytochrome P450 (CYP450) genes necessary for ecdysone biosynthesis (*Colombani et al., 2005*; *Layalle et al., 2008*). Increasing IIS/TOR activity in the PGs causes precocious ecdysone biosynthesis, precocious critical weight transitions, precocious metamorphosis and dramatic reductions in body size (*Caldwell et al., 2005*; *Colombani et al., 2005*; *Mirth et al., 2005*; *Layalle et al., 2008*). Reducing IIS/TOR activity in the PGs induces the opposite effects. However, the mechanisms through which the IIS/TOR pathway mediates these effects have been unclear.

Under well-fed conditions, insulin-like peptides (ILPs) are secreted into the insect blood or hemolymph (*Masumura et al., 2000*; *Ikeya et al., 2002*; *Rulifson et al., 2002*; *Geminard et al., 2009*). By binding to the Insulin Receptor (InR), ILPs activate IIS/TOR signaling in the target tissues (*Brogiolo et al., 2001*; *Ikeya et al., 2002*). Activating IIS/TOR signaling regulates a series of phosphokinases, including Akt. Akt, in turn, phosphorylates a negative regulator of growth, Forkhead Box class O transcription factor (FoxO), displacing it from the nucleus to the cytoplasm (*Junger et al., 2003*). In starved larvae, FoxO localizes in the nucleus where it acts on its targets, such as 4E-binding protein (4E-BP, also known as Thor), to suppress cell growth and division (*Puig et al., 2003*).

In mammalian cells, FoxO binds to several nuclear hormone receptors (NHRs), such as constitutive androstane receptor (CAR) and pregnane X receptor (PXR), to regulate CYP450 expression (*Kodama et al., 2004*). The functional ecdysone receptor is composed of two NHRs, Ecdysone Receptor (EcR) and Ultraspiracle (Usp). Since many CYP450 enzymes are involved in ecdysone biosynthesis (*Gilbert et al., 2002*), this led us to the hypothesis that the effect of IIS/TOR signaling on ecdysone biosynthesis is mediated by the interaction between FoxO and either EcR or Usp.

Here, we provide definitive evidence that critical weight results from the small nutrition-sensitive ecdysone peak early in the L3. Further, we report that IIS/TOR regulates the timing of ecdysone biosynthesis at critical weight via a novel mechanism, the direct association of FoxO and Usp. With these findings, we have constructed a detailed model of the molecular mechanisms underlying environmentally-sensitive ecdysone biosynthesis during critical weight, an event that ultimately determines the duration of the growth period and accordingly final body size.

## Results

### Starvation delays ecdysone biosynthesis and critical weight

Previous studies have shown that activating IIS/TOR signaling in the PGs induces early critical weight transitions, precocious ecdysone biosynthesis at wandering, and precocious metamorphosis (*Caldwell et al., 2005*; *Colombani et al., 2005*; *Mirth et al., 2005*; *Layalle et al., 2008*). This has led authors to propose that the small pulse of ecdysone early in the L3 (*Warren et al., 2006*) is nutrition-sensitive and induces critical weight in *Drosophila* (*Mirth and Riddiford, 2007*; *Koyama et al., 2013*; *Mirth et al., 2014*). However, these studies have not measured ecdysone concentrations with sufficient resolution early in the instar to show that ecdysone biosynthesis was delayed in starved pre-critical weight larvae. Therefore, we first examined whether this early ecdysone peak is delayed in starved larvae. In accordance with our hypothesis, we found that the small ecdysone peak that occurs around 10 hr after L3 ecdysis (AL3E) in well-fed larvae is suppressed in starved larvae, at least until 18 hr AL3E (*Figure 1A*). Thus, the timing of this early peak is indeed sensitive to nutrition.

Next we reasoned that if this early peak of ecdysone induced critical weight, feeding ecdysone to starved, pre-critical weight larvae should eliminate the delay in their development. To determine when wild type larvae reach critical weight, we starved carefully staged larvae of defined age classes on non-nutritive agar and measured the time it takes for them to reach pupariation from the onset of starvation. A hallmark of critical weight is that before it is attained starvation delays the onset of metamorphosis (*Beadle et al., 1938*; *Nijhout and Williams, 1974b*; *Mirth et al., 2005*; *Shingleton et al., 2005*; *Stieper et al., 2008*), whereas after critical weight larvae metamorphose early when starved. We estimate the age at critical weight using breakpoint analysis, which fits a bi-segmental linear regression to the relationship between age at starvation and time to pupariation, and calculates the age at critical weight as the inflection point where this relationship changes (*Stieper et al., 2008*; *Ghosh et al., 2013*; *Testa et al., 2013*). We then use the linear relationship between larval weight and larval age to convert the age at which larvae reach critical weight to the size at which larvae reach critical weight (*Figure 1—figure supplement 1B*). Finally, we repeated the analysis on 1000 bootstrap datasets to generate 95% confidence intervals for the age and size of larvae when they reach critical weight. Data and scripts for the analysis of size and age at critical weight, including the growth rate data, for all genotypes and treatments are available from the Dryad Digital Repository: 10.5061/dryad.75940 (*Koyama et al., 2014*).

Wild type larvae reached critical weight at 8.66 hr AL3E (*Figure 1B*, *Supplementary file 1*), correlating with the time when well-fed, wild type larvae show a peak of ecdysone (*Figure 1A*). When we added the active form of ecdysone, 20-hydroxyecdysone (20E), to the medium even the youngest larvae no longer delayed their onset of metamorphosis when starved (*Figure 1B*). Instead, larvae

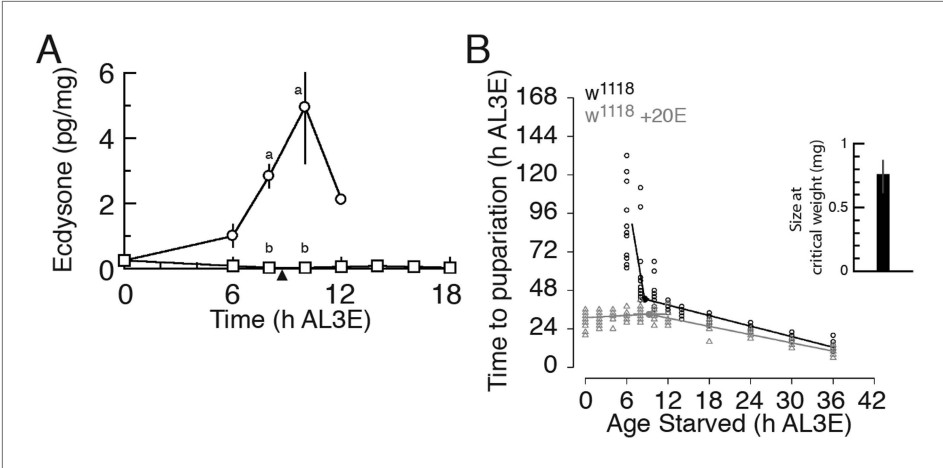

**Figure 1**. Nutrition regulates the timing of the critical weight ecdysone peak and exogenous ecdysone eliminates developmental delays in pre-critical weight larvae. (**A**) Nutrition is necessary to induce a small ecdysone peak at the early L3. We used 30–38 w[1118] larvae for each sample and three biologically independent samples for each time point. Each point indicates the mean ecdysone concentration ± SEM. Points sharing the same letter indicate the mean concentration at the time ±2 hr are statistically indistinguishable from one another; points that differ in letters are significantly different ($p < 0.05$). The arrowhead along the x axes indicates the age at which w[1118] larvae reached critical weight from *Figure 1B*. (**B**) Exogenous ecdysone administration throughout the L3 eliminates developmental delay in starved, pre-critical weight w[1118] larvae. The larvae were continuously fed a fly medium containing 0.15 mg/g 20E or transferred at given time points on to a starvation medium (1% agar) containing the same concentration of 20E. Inset shows the weight ±95% confidence intervals at which larvae reach critical weight. The age and size at which larvae reach critical weight was determined using breakpoint analysis and means and ±95% confidence intervals were calculated from 1000 bootstrap datasets.
The following figure supplement is available for figure 1:

**Figure supplement 1**. Ecdysone administration reduced body size.

starved on 20E-supplemented agar between the ages 0–8 hr AL3E pupariated 32 hr after the onset of starvation (*Supplementary file 1*). Finally, larvae fed 20E-supplemented fly medium throughout the L3 were more than 25% smaller than control larvae (*Figure 1—figure supplement 1A*). These results demonstrate that this early peak of ecdysone is nutrition sensitive and that it induces critical weight.

## The IIS/TOR and ecdysone pathways interact via FoxO–Usp association

We next sought to understand how nutrition regulated the timing of the critical weight ecdysone peak. We hypothesized that IIS/TOR signaling controlled the timing of this ecdysone peak, and therefore critical weight, via FoxO. We reasoned that if FoxO was involved in regulating ecdysone biosynthesis, FoxO would be present in the PG nuclei immediately after the molt to the L3 and would become progressively excluded from the nucleus as the larvae fed and approached critical weight. We found that FoxO was localized primarily in the nuclei of the PG cells of newly ecdysed L3 larvae (0 hr AL3E) (*Figure 2A*). As the larvae fed, FoxO was gradually transported out of the nuclei into the cytoplasm. At 5 hr AL3E, FoxO appeared evenly dispersed inside the PG cells (*Figure 2B*). By 10 hr AL3E, immediately after critical weight (*Figure 1B*), FoxO was mostly localized in cytoplasm of fed larvae (*Figure 2C,D*). Thus, FoxO appears to be progressively transported out of the nucleus as larvae approached critical weight.

For FoxO to regulate ecdysone biosynthesis in a nutrition-dependent manner, we would expect that it would remain in the nucleus in starved, pre-critical weight larvae. In larvae starved from 0–15 hr AL3E, FoxO remained in the nuclei of the PG cells (*Figure 2E*). In contrast, FoxO was found primarily in the cytoplasm in fed controls. Taken together, the localization of FoxO suggests that it could be involved in regulating ecdysone biosynthesis at critical weight.

Since FoxO associates with NHRs to regulate CYP450 gene expression in mammalian cells, we hypothesized that FoxO could associate with either EcR or Usp to regulate the nutrition-sensitive

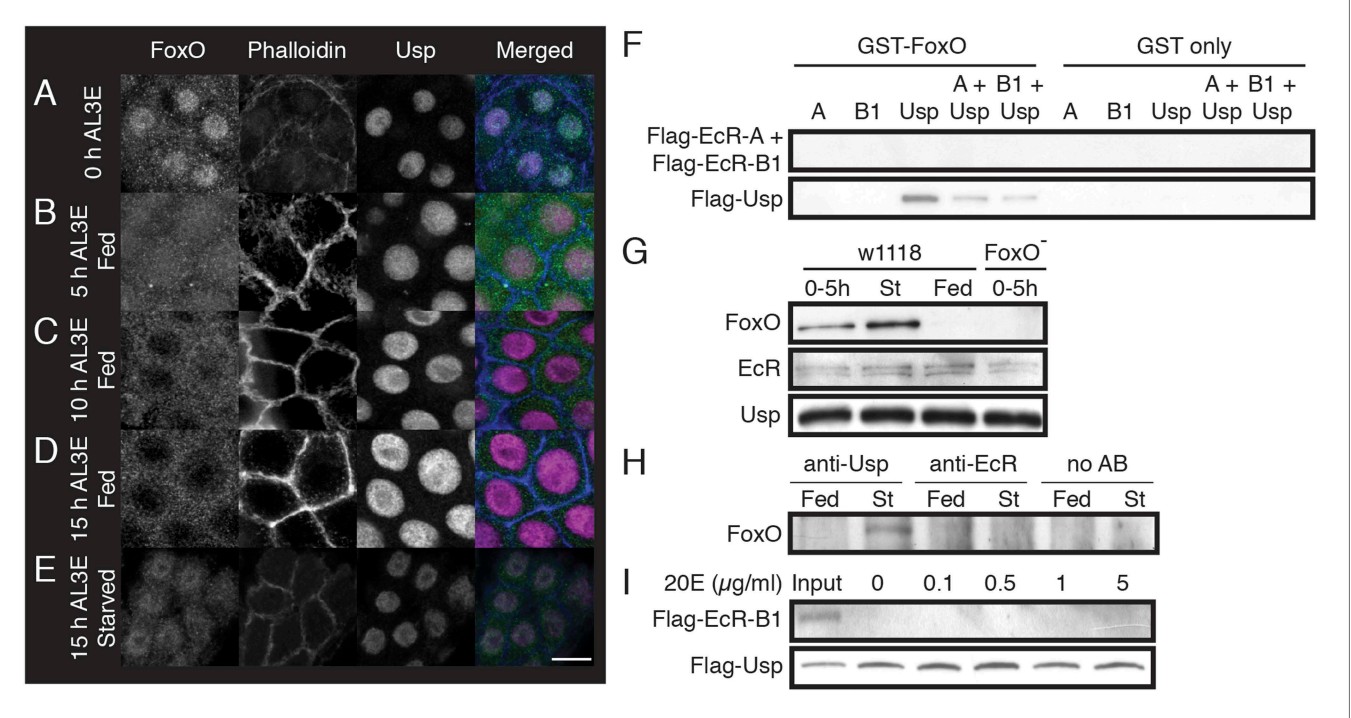

**Figure 2**. FoxO co-localizes with Usp in the PGs of pre-critical weight larvae and FoxO binds to Usp. (**A–E**) FoxO progressively moved out of the nuclei and into the cytoplasm of the PG cells in response to nutrition. PGs from w[1118] larvae at the onset of the L3 (**A**), fed for 5 (**B**), 10 (**C**) and 15 hr (**D**) or starved for 15 hr (**E**) were immunostained for FoxO, Usp and phalloidin. The scale bar is 10 μm. (**F**) GST-pulldown shows that FoxO binds to Usp but not to EcR. (**G**) FoxO associates with Usp before larvae reach critical weight but does not affect EcR–Usp association. Newly molted w[1118] larvae (0–5 hr AL3E) were either protein-starved (St) on 20% sucrose solution or fed on a standard fly medium (Fed) for additional 24 hr, and then the anterior halves of larvae without the fat body and salivary glands were used for protein extraction. We also examined pre-critical weight FoxO mutant (FoxO Δ94/Df(3R) Exel8159) larvae (0–5 hr) as a negative control. Precipitation was performed using the anti-Usp antibody. (**H**) Usp but not EcR associates with FoxO in co-immunoprecipitation assays using anti-Usp and anti-EcR antibodies. No AB indicates the no-antibody control. Protein extracts were prepared as in (**G**). (**I**) Presence of 20E neither changes FoxO–Usp binding properties nor induces FoxO–EcR association in a GST-pulldown assay.

ecdysone peak by regulating the expression of CYP450 ecdysone biosynthesis genes. Using GST-pulldown assays, we found that FoxO bound to Usp but not to EcR in vitro (**Figure 2F**). Co-immunoprecipitation experiments using larval extracts showed that FoxO bound to Usp only in pre-critical weight or starved larvae, but not in well-fed post-critical weight larvae (**Figure 2G,H**). FoxO neither bound to EcR, nor did it impede EcR/Usp binding in starved larvae (**Figure 2G,H**). This suggests that FoxO could interact with Usp to regulate the critical weight ecdysone peak, and further that this interaction is unlikely to interfere with EcR/Usp function.

Because in vertebrates FoxO-NHR interactions sometimes change in the presence of hormones (**Schuur et al., 2001**; **Li et al., 2003**; **Kodama et al., 2004**), we tested whether 20E altered FoxO/Usp binding or induced FoxO/EcR binding. The presence of 20E neither changed the binding properties of the FoxO–Usp interaction nor induced a FoxO–EcR association (**Figure 2I**).

## Both FoxO and Usp regulate critical weight

If FoxO/Usp interactions regulate ecdysone biosynthesis at critical weight, we would expect that altering the expression of FoxO or Usp in the PGs would change both the size and the age at which larvae reach critical weight. We used *Phantom* (*Phm*)-*Gal4*, a Gal4 driver specific for the PG cells, to overexpress FoxO in the PGs. These larvae attained critical weight at larger sizes and 10 hr later than in controls (**Figure 3A**). Overexpressing Usp in the PGs did not produce any significant difference in either the size or the age at which critical weight was achieved (**Figure 3B**). Overexpressing both FoxO and Usp in the PGs resulted in larvae that reached critical weight more than 13 hr later and about 1 mg larger than control larvae (**Figure 3C**). Further, the size of these larvae at critical weight

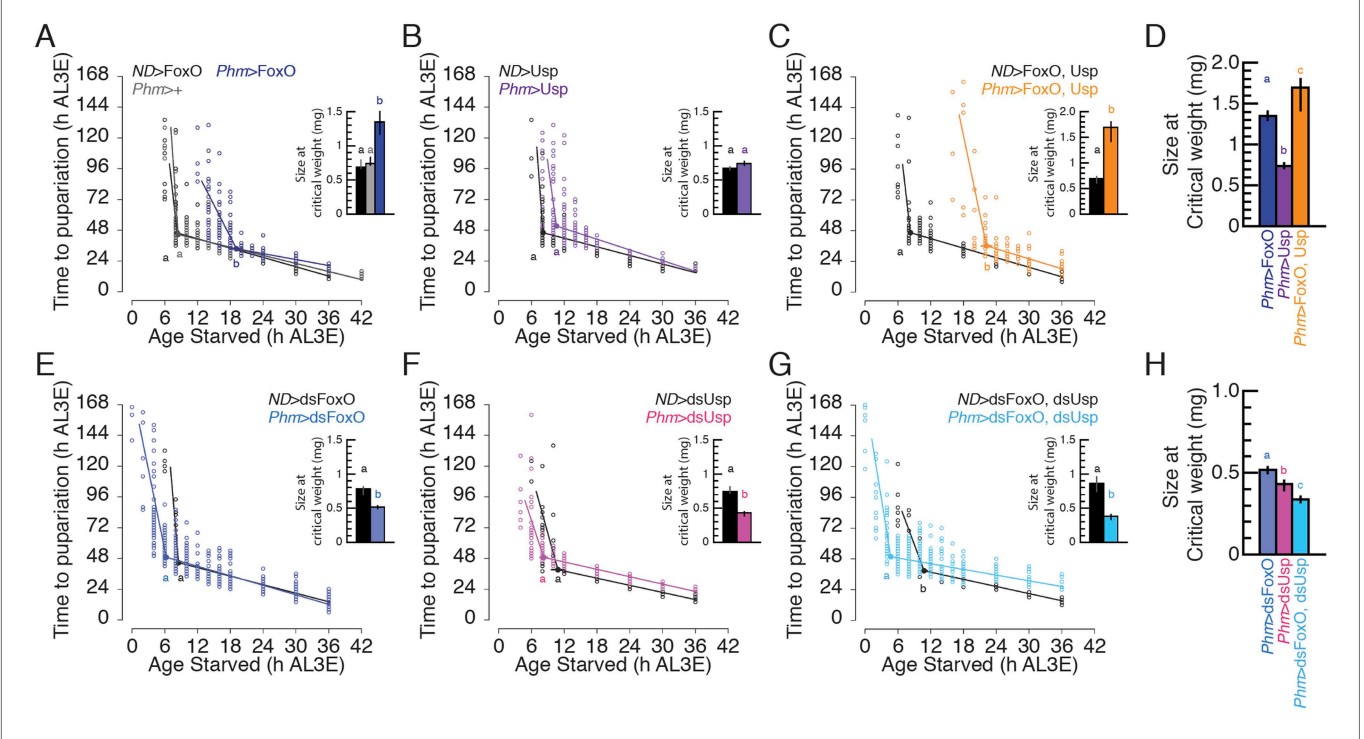

**Figure 3**. Manipulating FoxO and/or Usp in the PGs changes the timing of critical weight. (**A–C**) Age at which animals are starved in relation to the time to pupariation from the onset of starvation for *Phm>FoxO* (**A**), *Phm>Usp* (**B**), *Phm>FoxO, Usp* (**C**) animals and their parental controls (*Phm>+* and *no driver*, *ND*). (**D**) Critical weight was compared when either or both FoxO and/or Usp were overexpressed in the PGs. (**E–G**) Age at which animals are starved in relation to the time to pupariation from the onset of starvation for *Phm>dsFoxO* (**E**), *Phm>dsUsp* (**F**), and *Phm>dsFoxO, dsUsp* (**G**) and their parental controls. (**H**) Critical weight was compared when either or both FoxO and/or Usp were knocked down in the PGs. Insets show the size at critical weight ±95% confidence intervals. The age at which larvae reached critical weight ±95% confidence intervals was determined by breakpoint analysis. Points or columns sharing the same letter indicate the groups that are statistically indistinguishable from one another; points or columns that differ in letters are significantly different (Permutation Test, p < 0.05).

The following figure supplement is available for figure 3:

**Figure supplement 1**. Manipulating FoxO and/or Usp in the PGs changes the body size.

was significantly larger than when either FoxO or Usp was overexpressed in the PGs alone (***Figure 3D***). These data suggest that both FoxO and Usp regulate the timing of critical weight.

In contrast, knocking down either FoxO or Usp alone in the PGs reduced the size but not the age at critical weight (***Figure 3E*** or ***Figure 3F***, respectively). When we simultaneously knocked down both FoxO and Usp in the PGs, larvae reached critical weight significantly earlier at smaller sizes (***Figure 3G***) than knocking down either FoxO or Usp alone (***Figure 3H***). These knock down experiments corroborate our results from our FoxO and Usp overexpression experiments and provide further evidence that both FoxO and Usp suppress ecdysone biosynthesis.

Because critical weight is a key determinant of final body size, we also weighed pharate adults as a proxy of final adult size. Overexpressing either FoxO or Usp in the PGs significantly increased body size compared to parental controls (***Figure 3—figure supplement 1A***). In addition, females that overexpressed both FoxO and Usp together in the PGs had significantly larger body sizes than those overexpressing either FoxO or Usp alone (***Figure 3—figure supplement 1A***, ANOVA interaction term, p = 0.045). Knocking down Usp resulted in a significant decrease in body size (***Figure 3—figure supplement 1B***). Knocking down FoxO caused a slight, but significant decrease in body size in males but not in females. However, knocking down both Usp and FoxO in the PGs dramatically reduced body size (***Figure 3—figure supplement 1B,C***). These results demonstrate that altering the size and timing of critical weight, by manipulating expression of FoxO and Usp, has definitive effects on final adult body size.

## FoxO and Usp regulate *phantom, disembodied* and *e74B* gene expression

Our data show that the earliest ecdysone peak in the L3 regulates critical weight and that FoxO and Usp alter the timing of this transition. To confirm that FoxO and Usp regulate critical weight by controlling the timing of ecdysone biosynthesis, we examined the expression of two CYP450 ecdysone biosynthetic genes, *phm* and *disembodied* (*dib*), known to be sensitive to IIS/TOR signaling (**Colombani et al., 2005**; **Layalle et al., 2008**), in larvae with altered FoxO and Usp expression. In addition, we quantified the expression of an ecdysone response gene, *e74B* (*eip74ef isoform B*), which tracks the early effects of ecdysone signaling (**Caldwell et al., 2005**; **Colombani et al., 2005**; **Layalle et al., 2008**) in these larvae. In the parental controls, both *phm* and *dib* increased in expression around 8 hr AL3E, shortly before the critical weight ecdysone peak (**Figure 4A,B,D,E**). *E74B* expression peaks around 12 hr in parental controls, after the critical weight ecdysone peak (**Figure 4C,F**). When both FoxO and Usp were overexpressed in the PGs, the increase in *phm* and *dib* expression was delayed (**Figure 4A,B**) and *e74B* expression remained low up to 20 hr AL3E (**Figure 4C**). In contrast, when we

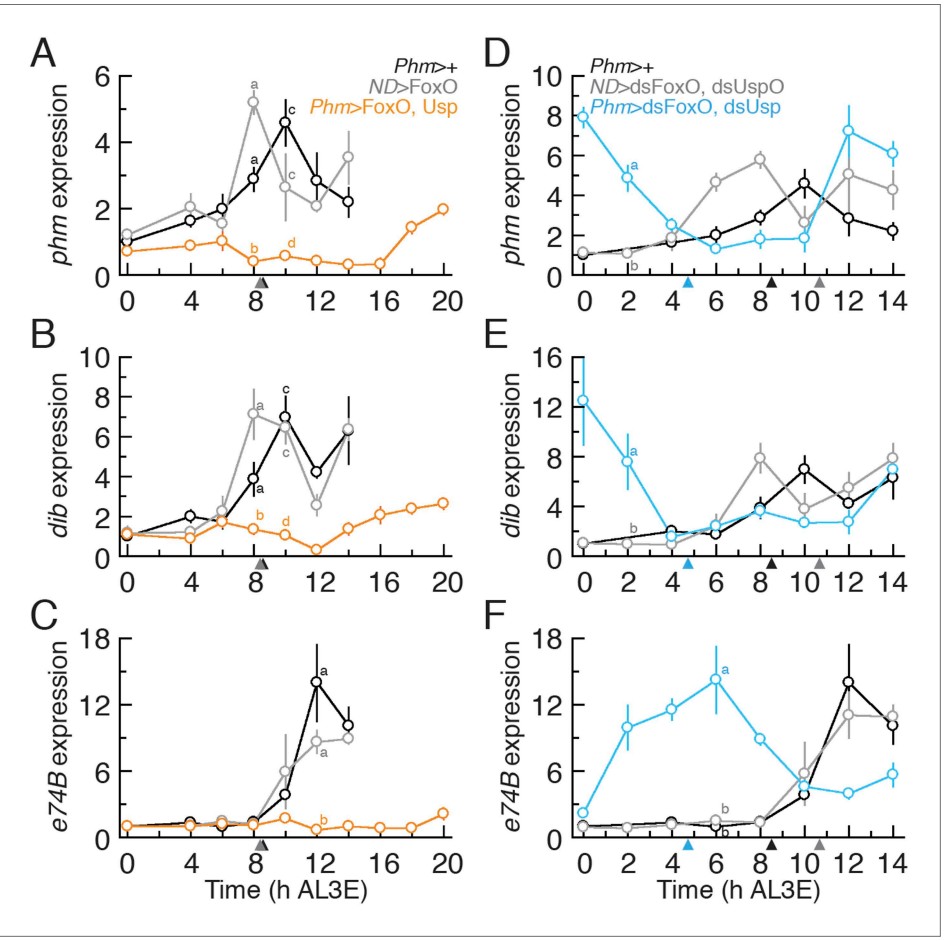

**Figure 4**. Altering FoxO and Usp expression also alters *phm*, *dib* and *e74B* expression. (**A–C**) Relative *phm* (**A**), *dib* (**B**) and *e74B* (**C**) mRNA expression in *Phm>FoxO, Usp* animals were quantified by quantitative PCR. (**D–F**) Relative *phm* (**D**), *dib* (**E**) and *e74B* (**F**) mRNA expression in *Phm>dsFoxO, dsUsp* animals were quantified by qPCR. We normalized the values using an internal control, *RpL3*. Then, we standardized the expression level of each gene by fixing the values at 0 hr in *Phm>+* animals as 1 in all figures. We used 4–6 larvae for each sample and three biologically independent samples for each time point. Each point indicates the relative mean expression ± SEM. Points sharing the same letter indicate the mean expression at the time ±2 hr are statistically indistinguishable from one another; points that differ in letters are significantly different (p < 0.05). Arrowheads along the x axes indicate the age at which each genotype reached critical weight from **Figure 3A,C,G**.

knocked down FoxO and Usp, both *phm* and *dib* expression levels were high immediately after the molt to the L3 (*Figure 4D,E*) and *e74B* expression was nearly undetectable at ecdysis but increased rapidly thereafter (*Figure 4F*). Taken together, these results suggest that alterations in FoxO and Usp affect the timing of ecdysone biosynthesis at critical weight.

## Identifying the binding sites for FoxO–Usp interactions

Although our results suggest that both FoxO and Usp act in the PGs to regulate the timing of critical weight ecdysone peak, thereby mediating the timing of critical weight, they do not allow us to distinguish whether FoxO and Usp regulate ecdysone biosynthesis independently or together via the FoxO/Usp complex. To discern between these two possibilities, we developed a genetic tool to manipulate the FoxO–Usp interaction.

First, we identified the Usp binding site in the FoxO protein using GST-pulldown assays. We created overlapping GST-tagged FoxO fragments and, using increasingly smaller overlapping fragments, we narrowed down the Usp binding region to a 35 amino acid region overlapping with 5 amino acids in the C-terminal end of the forkhead (DNA binding) domain (*Figure 5A*). This motif is well conserved across arthropod species including ticks and water fleas, but is not conserved in FoxO proteins in other ecdysozoans or vertebrates (*Figure 5—figure supplement 1*). Interestingly, this Usp binding motif is different from the well-known 'LXXLL'-type NHR binding motif identified in vertebrates (*Heery et al., 1997*). Next, we identified eighteen candidate amino acids by comparing the crystal structure of mammalian FoxO3a to the *Drosophila* FoxO sequence (*Tsai et al., 2007*) and selecting residues that occupied positions permissive for protein–protein interactions. We mutated each of these to alanine. At least 4 of the 18 amino acid residues appeared to be involved in FoxO–Usp binding (residues W172, N175, R202 and K204). When we introduced these single point mutations into the full length FoxO protein, they showed only mild reductions in FoxO–Usp binding (*Figure 5A*). We then tested four double–mutant combinations (W172-R202, W172-K204, N175-R202, and N175-K204) all of which were sufficient to dramatically reduce FoxO–Usp interactions (*Figure 5A*). Two of these double–mutant combinations partially reduced the FoxO activity (W172-R202 and W172-K204) (*Figure 5—figure supplement 2C,D*), as determined by the expression of known FoxO targets InR and 4E-BP (*Puig et al., 2003*). Because our aim was to disrupt FoxO–Usp binding, but not FoxO function, these were excluded from further analyses. The remaining two double–mutant combinations (N175-R202 or FoxO NR, and

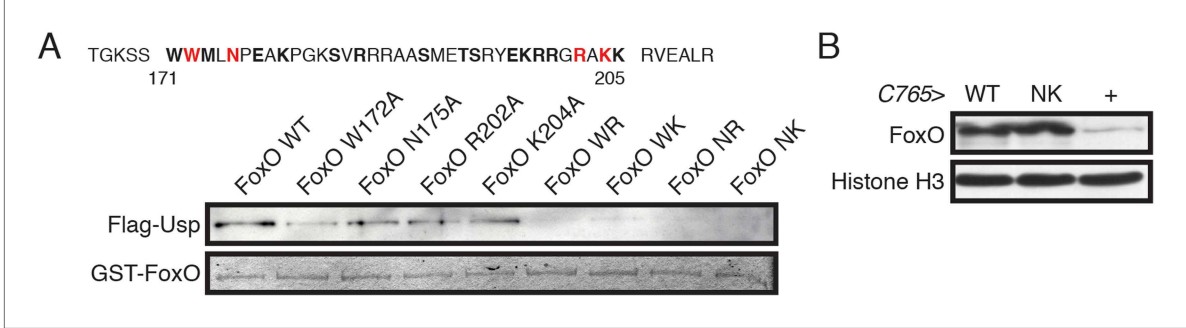

**Figure 5**. The Usp binding site in FoxO protein was identified and FoxO NK mutation showed reduced binding affinity to Usp. (**A**) Point mutations were induced in the FoxO protein at site of the amino acids indicated in bold. Point mutations indicated in red showed reduced binding affinity to Usp. For a loading control, we used Coomassie Brilliant Blue staining to detect GST-FoxO fusion protein. (**B**) UAS FoxO and UAS FoxO NK transgenes show similar expression levels. We overexpressed either FoxO or FoxO NK using C765-*Gal4*. The wing discs were dissected from early white prepupae. We used C765>+ as a parental control, and Histone H3 as a loading control.

The following figure supplements are available for figure 5:

**Figure supplement 1**. Amino acid sequence alignments of the Usp binding motif across arthropods and with non-arthropods.

**Figure supplement 2**. FoxO NK does not change the Usp-independent function of FoxO.

**Figure supplement 3**. FoxO NK shows Usp-independent FoxO activity in transgenic flies.

N175-K204 or FoxO NK) showed normal translocation to the nucleus (*Figure 5—figure supplement 2A,B*) and did not affect FoxO's ability to regulate InR and 4E-BP promoter activities (*Figure 5—figure supplement 2C,D*, respectively).

Finally, we tested whether FoxO NR and FoxO NK retained the ability to suppress tissue growth. We placed the wild type FoxO, FoxO NR and FoxO NK constructs under the control of a UAS promoter and inserted them into the fly genome, using targeted integration (*Groth et al., 2004*) to control for positional effects of the transgenes. All three constructs have the same genetic background and differ only in the two amino acids mutated to interfere with FoxO–Usp binding. We then drove expression of the FoxO, FoxO NR and FoxO NK in the wings, using the C765- and *MS1096-Gal4* drivers, and in the eyes, using the *eyeless (Ey)-* and *GMR-Gal4* drivers. For all drivers, overexpression of FoxO and FoxO NK reduced organ size to a similar degree (*Figure 5—figure supplement 3*). FoxO NR showed milder reductions in tissue size (*Figure 5—figure supplement 3*), therefore in subsequent experiments we used FoxO NK. We confirmed that both FoxO and FoxO NK expressed FoxO protein at the same level (*Figure 5B*). Taken together, FoxO NK showed reduced FoxO–Usp affinity, but maintained Usp-independent FoxO function.

## The FoxO/Usp complex suppresses critical weight

To explore whether FoxO and Usp regulate critical weight independently or together as a complex, we drove expression of FoxO NK in the PGs of developing larvae and compared them with larvae expressing wild type FoxO in the PGs. Larvae that overexpressed FoxO NK in their PGs reached critical weight earlier and at smaller sizes than those that overexpressed wild type FoxO, albeit later than the parental controls (*Figure 6A*). Thus, impeding FoxO–Usp binding reduced the delay in critical weight induced by FoxO overexpression. Similarly, pupae in which FoxO NK was overexpressed in the PGs were significantly smaller than pupae that overexpressed wild type FoxO, although they were still larger than pupae from the parental controls (*Figure 6—figure supplement 1A*).

Because FoxO alters developmental timing via its effects in other tissues, like the fat body (*Colombani et al., 2005*), we eliminated effects of the endogenous gene by overexpressing FoxO in the PGs of *FoxO null* mutant larvae (*FoxO Δ94/Df(3R)Exel8159*) (*Slack et al., 2011*). When we used the *Phm-Gal4* driver to overexpress FoxO in the PGs of *FoxO null* animals, most larvae did not survive to the L3. To circumvent this problem, we used *P0206-Gal4*, which expresses Gal4 moderately in the PGs (*Mirth et al., 2005*). Since *P0206-Gal4* driver expresses *Gal4* in other tissues such as the oenocytes and corpora allata, we tested the effects of overexpressing FoxO in these other tissues. To do this, we compared the duration of the L3 and final body size when overexpressing FoxO in the ring gland, oenocytes and corpora allata, using *P0206-Gal4*, in the oenocytes alone, using *PromE(800)-Gal4* (*Billeter et al., 2009*), and in the corpora allata alone, using *Aug21-Gal4* (*Mirth et al., 2005*) all in the *FoxO null* mutant background. We found that overexpressing FoxO in the oenocytes and corpora allata does not affect the duration of the L3. In contrast, overexpressing FoxO in the ring glands using *P0206-Gal4* prolonged the duration of the L3 compared to parental controls (*Figure 6—figure supplement 2A*). Further, overexpressing FoxO in the oenocytes and corpora allata did not affect final body size, whereas overexpressing FoxO using *P0206-Gal4* increased body size (*Figure 6—figure supplement 2B*). Taken together, our data suggest that FoxO overexpression in the oenocytes and corpora allata had no measurable effect on growth rate or the duration of the L3. Thus, we conclude that developmental delay and size increase induced by overexpressing FoxO using *P0206-Gal4* is due to the functions of FoxO in the PGs.

Similar to what we found in the *FoxO* wild type background, *FoxO null* larvae that overexpressed FoxO NK in their PGs reached critical weight earlier and at smaller sizes than larvae that overexpressed wild type FoxO in their PGs (*Figure 6B*). Further, we confirmed that overexpressing both FoxO NK and Usp in the PGs of *FoxO null* mutant larvae did not alter the age and size at critical weight (*Figure 6—figure supplement 2C*) nor did it alter final body size when compared to overexpressing FoxO NK alone (*Figure 6—figure supplement 2D*). These results suggest that Usp does not affect the timing of critical weight on its own and that critical weight is regulated, at least in part, by FoxO/Usp.

## Feeding ecdysone is sufficient to eliminate the developmental delays induced by FoxO overexpression in the PGs

We next tested whether exogenous ecdysone could rescue the delay in critical weight induced by the FoxO/Usp complex. To do this, we assessed critical weight in *P0206>FoxO, FoxO null* larvae and the

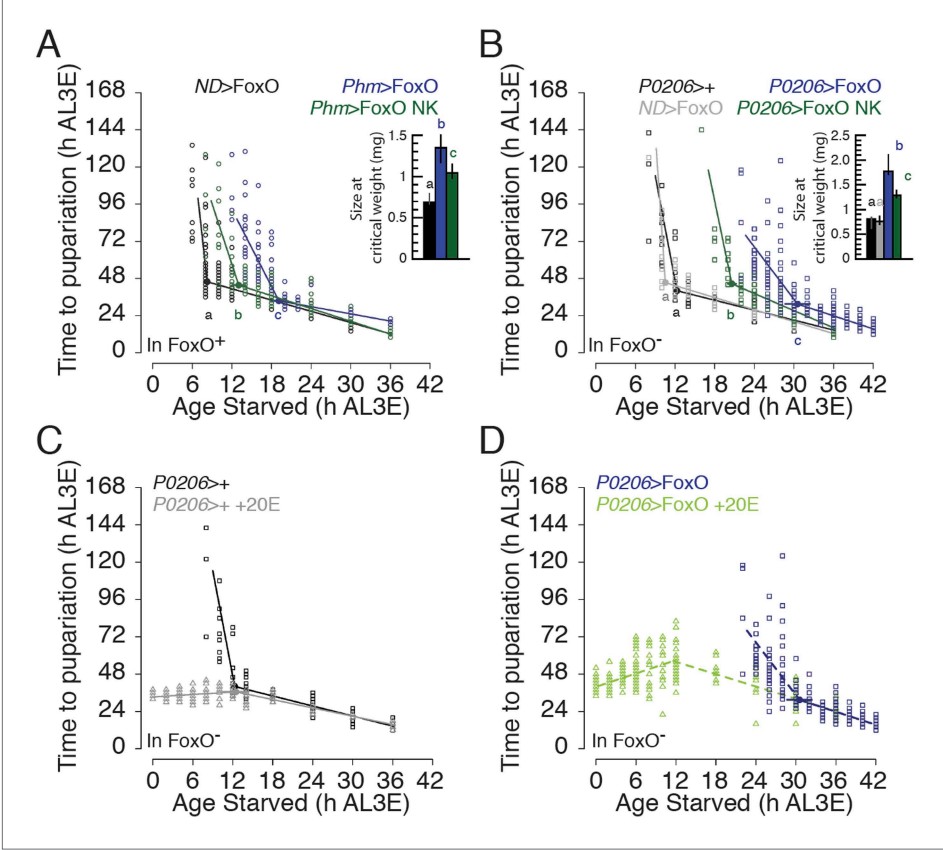

**Figure 6**. Interfering FoxO–Usp association changes the timing of critical weight. (**A** and **B**) Age at which animals are starved in relation to the time to pupariation from the onset of starvation for *Phm*>FoxO and *Phm*>FoxO NK in the FoxO wild type background (**A**), and *P0206*>FoxO and *P0206*>FoxO NK in the FoxO mutant background (**B**) and their parental controls. (**C** and **D**) Feeding ecdysone throughout the L3 eliminates developmental delay in *P0206*>+ (**C**) and in *P0206*>FoxO (**D**), FoxO mutant larvae. The larvae were continuously fed 0.15 mg/g 20E as described in *Figure 1*. Data for *ND*>FoxO and *Phm*>FoxO in **A** and for the non-20E-treated data in **C** and **D** were re-plotted from *Figure 3A* and *Figure 6B*, respectively. Insets show the size at critical weight (mg) ±95% confidence intervals. The age at which larvae reach critical weight ±95% confidence intervals was determined by breakpoint analysis. Points or columns sharing the same letters indicate the groups that are statistically indistinguishable from one another; points or columns that differ in letters are significantly different (Permutation Test, p < 0.05).

The following figure supplements are available for figure 6:

**Figure supplement 1**. FoxO NK overexpression in the PGs reduced the body size phenotype.

**Figure supplement 2**. The effects of overexpressing FoxO using *P0206-Gal4* is due to the function of FoxO in the PGs and FoxO NK shows proper Usp-independent transcriptional activity.

---

*P0206*>+, *FoxO null* parental controls on medium supplemented with 20E. We found that adding 20E to the medium altered the relationship between age at starvation and time to the onset of metamorphosis in both genotypes (*Figure 6D*). In the parental controls, larvae starved before 12 hr AL3E on ecdysone-supplemented agar did not delay the onset of metamorphosis, pupariating 36 hr after starvation (*Figure 6C*, *Supplementary file 1*). When we starved *P0206*>FoxO, *FoxO null* larvae on 20E supplemented agar, time to pupariation increased with age of starvation from 0–12 hr AL3E, with larvae showing a maximum time to pupariation of 57 hr (*Supplementary file 1*) at 12 hr AL3E, then decreased thereafter. This suggests that (1) 20E administration eliminated the strong delays in time to metamorphosis seen in pre-critical weight *P0206*>FoxO, *FoxO null* larvae and (2) FoxO overexpression in the PGs has additional stage-specific effects on the time to pupariation after the critical weight ecdysone peak and this effect is nutrition-sensitive.

In addition to the effects we observed on developmental time, we found that both *P0206>FoxO, FoxO null* and *P0206, FoxO null* parental control animals were significantly smaller when they were continuously fed 20E-supplemented normal fly medium when compared to animals reared on normal fly medium alone (*Figure 6—figure supplement 1C,D*). These data demonstrate that altering the timing of critical weight, by exogenous ecdysone administration, impacts final adult size.

## The FoxO/Usp complex suppresses ecdysone biosynthesis

The goal of this study was to uncover the molecular mechanism through which nutrition regulates ecdysone synthesis at critical weight. Our results show that FoxO and Usp interact to regulate critical weight and suggest that this interaction alters the timing of the ecdysone peak. To definitively test whether the FoxO/Usp complex regulates ecdysone biosynthesis at critical weight, we examined whether the FoxO/Usp complex altered the timing of *phm* and *dib* mRNA expression, the timing of *e74B* expression and finally the timing of the ecdysone peak itself.

The expression of both *phm* and *dib* mRNA peaked shortly before critical weight in the parental controls (*Figure 7A,B* and *Figure 7—figure supplement 1A–C*). However, *P0206>FoxO, FoxO null* larvae showed significant delays in this peak (*Figure 7B,D*). Overexpressing FoxO NK in the PGs reduced the delays induced by FoxO overexpression (*Figure 7B,D*). Similarly in the wild type background, *phm* and *dib* expression was upregulated significantly earlier when FoxO NK was expressed in the PGs than when FoxO was overexpressed in this tissue (*Figure 7—figure supplement 1A,B*).

Even if we observe alterations in ecdysone biosynthesis gene expression, this does not necessarily mean that ecdysone biosynthesis is affected when we manipulate FoxO expression in the PGs. To assess if overexpressing FoxO in the PGs affected ecdysone signaling, we examined the expression of *e74B*. In the parental control larvae, *e74B* mRNA expression was up-regulated around 12 hr AL3E, shortly after critical weight (*Figure 7C*, *Figure 7—figure supplement 1C*). Overexpressing FoxO in the PGs delayed the up-regulation of *e74B* in both the *FoxO* mutant and wild type backgrounds (*Figure 7C*, *Figure 7—figure supplement 1C*). Finally, interfering with FoxO/Usp complexes, by overexpressing FoxO NK, in the PGs reduced this delay in both the *FoxO* mutant and wild type backgrounds (*Figure 7C*, *Figure 7—figure supplement 1C*). Thus, FoxO/Usp complex plays a role in regulating the dynamics of ecdysone signaling at critical weight.

Finally, to show that the FoxO/Usp complex regulates ecdysone biosynthesis at critical weight, we measured ecdysone concentrations in larvae that expressed either FoxO or FoxO NK in their PGs from 6 hr AL3E until the nutrition-dependent critical weight ecdysone peak. Overexpressing FoxO in the PGs of FoxO mutant larvae induced a significant delay in the critical weight ecdysone peak (*Figure 7D*). In addition, the maximum concentration of this peak was approximately 50% lower than the critical weight ecdysone peak in parental controls. The critical weight ecdysone peak occurred significantly earlier in *P0206>FoxO NK* larvae than in *P0206>FoxO* larvae. This difference in the timing of the critical weight ecdysone peak was not due to differences in the effects between FoxO and FoxO NK on PG size. Overexpressing either FoxO or FoxO NK induced indistinguishable reductions in PG size (*Figure 7E*). Taken together our results show that FoxO acts to control the timing of ecdysone biosynthesis via the FoxO/Usp complex, but also via Usp-independent mechanisms.

## Discussion

Environmental conditions influence developmental processes by affecting hormone synthesis in many organisms. These interactions form the basis of developmental plasticity, and can act to resize and reshape the whole animal. Although environmental effects on hormone synthesis have been identified as a mechanism underlying plasticity in many insects, what causes hormones to become environmentally-sensitive was poorly understood. Here, we demonstrated that FoxO associates with Usp to regulate nutrition-sensitive ecdysone biosynthesis. Our work uncovers a novel mechanism that allows hormone biosynthesis to become environmentally-sensitive at key developmental events, in this case to control plasticity in body size.

### The FoxO/Usp complex regulates critical weight by regulating ecdysone biosynthesis

Increasing IIS/TOR activity in the PGs induces precocious critical weight and reducing its activity in the PGs prolongs this transition (*Caldwell et al., 2005*; *Colombani et al., 2005*; *Mirth et al., 2005*; *Layalle et al., 2008*). Because IIS/TOR signaling positively regulates the expression of CYP450

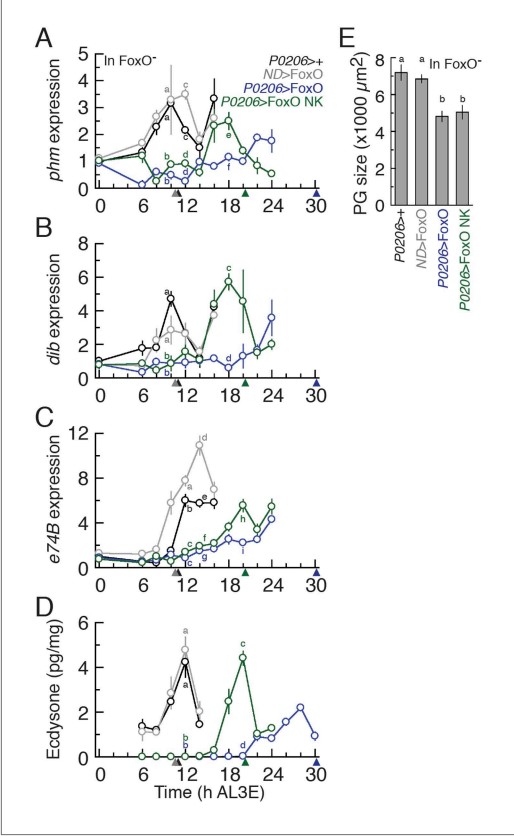

**Figure 7**. The FoxO/Usp complex suppresses critical weight through inhibiting ecdysone biosynthesis in the PGs. (**A–C**) Relative *phm* (**A**), *dib* (**B**), and the ecdysone response gene *e74B* mRNA expression (**C**) in the FoxO mutant backgrounds were quantified by qPCR. We normalized the values by an internal control, *ribosomal protein large subunit 3* (*RpL3*). Then, we standardized the expression level of each gene by fixing the values at 0 hr in *P0206>+* animals as 1. We used 5–6 larvae for each sample and three biologically independent samples for each time point. Each point indicates the relative mean expression ± SEM. Points sharing the same letters indicate the mean expression at the time ±2 hr are statistically indistinguishable from one another; points that differ in letters are significantly different (p < 0.05). (**D**) The FoxO/Usp complex suppresses ecdysone biosynthesis during critical weight period in larvae with FoxO mutant backgrounds. We used 32–46 larvae for each sample and three biologically independent samples for each time point. Each point indicates the mean ecdysone concentration ± SEM. Points sharing the same letter indicate the mean concentration at the time ±2 hr are statistically indistinguishable from one another; points that differ in letters are significantly different (p < 0.05). Arrowheads along the x axes indicate the age at which each genotype reached critical weight from *Figure 6B*. (**E**) Overexpressing FoxO or FoxO NK equally reduces the PG size of the *FoxO null* mutant larvae. The PGs were dissected at 24 hr AL3E

*Figure 7. Continued on next page*

ecdysone biosynthetic genes, *phm* and *dib* (*Colombani et al., 2005*; *Layalle et al., 2008*), we previously hypothesized that IIS/TOR exerted these effects by regulating the timing of the small peak of ecdysone that coincides with critical weight (*Mirth and Riddiford, 2007*; *Mirth and Shingleton, 2012*).

Our data both tested this hypothesis and identified a novel interaction between the IIS/TOR and ecdysone signaling pathways. We have found that interactions between FoxO and Usp regulate ecdysone biosynthesis, critical weight and body size. This allows us to propose a model for nutrition-sensitive ecdysone biosynthesis during critical weight (*Figure 8*). During the molt to the L3, larvae undergo a period of starvation while they expel their mouthparts (*Park et al., 2002*, *2003*). As a consequence, IIS/TOR signaling activity in the PGs is reduced and FoxO remains in the nucleus and forms a complex with Usp. The FoxO/Usp complex suppresses ecdysone biosynthesis at least in part by repressing transcription of *phm* and *dib,* although we do not know whether this repression is direct. Once larvae start feeding, increasing IIS/TOR activity in the PGs results in the phosphorylation of FoxO, causing the dissociation FoxO/Usp complexes as FoxO moves out of the nucleus. This progressive dissociation of FoxO/Usp complexes results in a gradual rise in ecdysone biosynthesis. Once ecdysone reaches a threshold, it triggers critical weight. Afterwards, the time to metamorphosis is set and can no longer be delayed by starvation. For other ecdysone peaks, a negative feedback loop induced by ecdysone signaling itself down-regulates ecdysone biosynthesis (*Sakurai and Williams, 1989*; *Takaki and Sakurai, 2003*; *Moeller et al., 2013*). We expect that negative feedback by ecdysone results in the decline in ecdysone biosynthesis after the critical weight peak.

In contrast, when larvae are starved before attaining critical weight, FoxO remains in the nucleus. In these larvae, the FoxO/Usp complex suppresses ecdysone biosynthesis and delays critical weight. Consequentially, the onset of metamorphosis is delayed. This work uncovers a mechanism that allows IIS/TOR signaling to control ecdysone biosynthesis, providing an elegant means for nutrition to regulate body size.

## Other regulators of ecdysone biosynthesis

Although the ecdysone peak at critical weight is environmentally-sensitive, many other peaks that occur throughout the larval period show less plasticity in response to environmental cues. Ecdysone

*Figure 7. Continued*

and stained with phalloidin. After photographing, these areas were quantified using the ImageJ. Each bar indicates the mean area ± SEM. *N* = 7–10. Columns sharing the same letter indicate the groups that are statistically indistinguishable from one another; columns that differ in letters are significantly different (p < 0.05).
The following figure supplement is available for figure 7:

**Figure supplement 1**. The FoxO/Usp complex delays ecdysone synthesis and ecdysone response gene expression in the FoxO wild type background.

biosynthesis is also regulated by a developmental neuropeptide, prothoracicotropic hormone (PTTH). Several extrinsic and intrinsic stimuli affect PTTH secretion, such as photoperiod, oxygen concentrations, signals released from damaged imaginal discs, and the sesquiterpenoid 'status quo' hormone juvenile hormone (*Truman, 1972*; *Nijhout and Williams, 1974a, 1974b*; *McBrayer et al., 2007*; *Halme et al., 2010*; *Callier and Nijhout, 2011*; *Colombani et al., 2012*; *Garelli et al., 2012*; *Mirth and Shingleton, 2012*). Activating downstream targets of PTTH signaling in the PGs accelerates the onset of metamorphosis (*Caldwell et al., 2005*) and ablating the PTTH-producing cells delays critical weight (*McBrayer et al., 2007*;

*Rewitz et al., 2009*). Further, without PTTH the ecdysone peak that stimulates wandering behavior, where the larvae emerge from the food to begin metamorphosis, is dramatically delayed (*McBrayer et al., 2007*; *Rewitz et al., 2009*; *Gibbens et al., 2011*). Thus in contrast to IIS/TOR signaling whose major effects are to control the critical weight ecdysone peak, PTTH regulates all ecdysone peaks. Why particular ecdysone peaks are more sensitive to IIS/TOR signaling is unclear, however understanding the mechanisms underlying this differential sensitivity may be key to understanding developmental plasticity.

## Usp-independent effects of FoxO on ecdysone biosynthesis

FoxO also regulates the critical weight ecdysone peak independently of Usp; overexpressing FoxO NK in the PGs still induces delays in ecdysone biosynthesis and critical weight, even if these delays are more moderate than those induced by wild type FoxO. Thus, our data suggest that FoxO plays additional roles in regulating ecdysone biosynthesis, either on its own or through interaction with other binding partners.

The effects of starvation on ecdysone biosynthesis do not appear to be the same for all stages of development. Even though starvation causes a delay in development before attaining critical weight, once they reach critical weight, starvation induces moderate acceleration in the time to metamorphosis (*Mirth et al., 2005*; *Stieper et al., 2008*). This suggests that reducing IIS/TOR signaling induces a mild acceleration of ecdysone biosynthesis at later stages of the L3 development. How IIS/TOR activity regulates ecdysone biosynthesis differently depending on the stage of development is unclear, but it may result from interaction of alternate FoxO binding partners.

## FoxO-NHR complexes and steroid hormone signaling

Our findings have broad implications for our understanding of the mechanisms of size regulation and the development of other environmentally-sensitive traits. In other insects, traits such as seasonal wing morphs in butterflies (*Koch and Bückmann, 1987*; *Koch et al., 1996*; *Oostra et al., 2011*) or horn length in male dung beetles (*Emlen and Nijhout, 1999*) arise from differential regulation of ecdysone biosynthesis (*Koyama et al., 2013*). Horn length in dung beetles is highly nutrition-dependent, with small, poorly-fed males bearing small horns and large, well-fed males having disproportionately larger horns (*Emlen, 1994, 1997*). Small-horned males have a characteristic peak of ecdysone in their final instar absent in their well-fed, larger conspecifics (*Emlen and Nijhout, 1999, 2001*). Our data propose a mechanism through which nutrition, via FoxO–Usp interactions, might regulate this peak (*Koyama et al., 2013*).

FoxO is also known to form complexes with many vertebrate NHRs, including thyroid hormone (*Zhao et al., 2001*), androgen (*Li et al., 2003*; *Fan et al., 2007*) and estrogen receptors (*Schuur et al., 2001*). The steroid sex hormones, such as testosterone and estrogen, are important for initiating puberty and the development of adult characters in humans. In girls, reaching a body mass of 48 kg determines the timing of first menses (*Frisch and Revelle, 1970*; *Freedman et al., 2003*; *Gluckman and Hanson, 2006*; *Ahmed et al., 2009*). Obese girls reach this mass faster, resulting in earlier onset of puberty (*Freedman et al., 2003*; *Gluckman and Hanson, 2006*; *Ahmed et al., 2009*) possibly due to higher levels of insulin signaling (*Codner and Cassorla, 2009*; *Lee et al., 2011*;

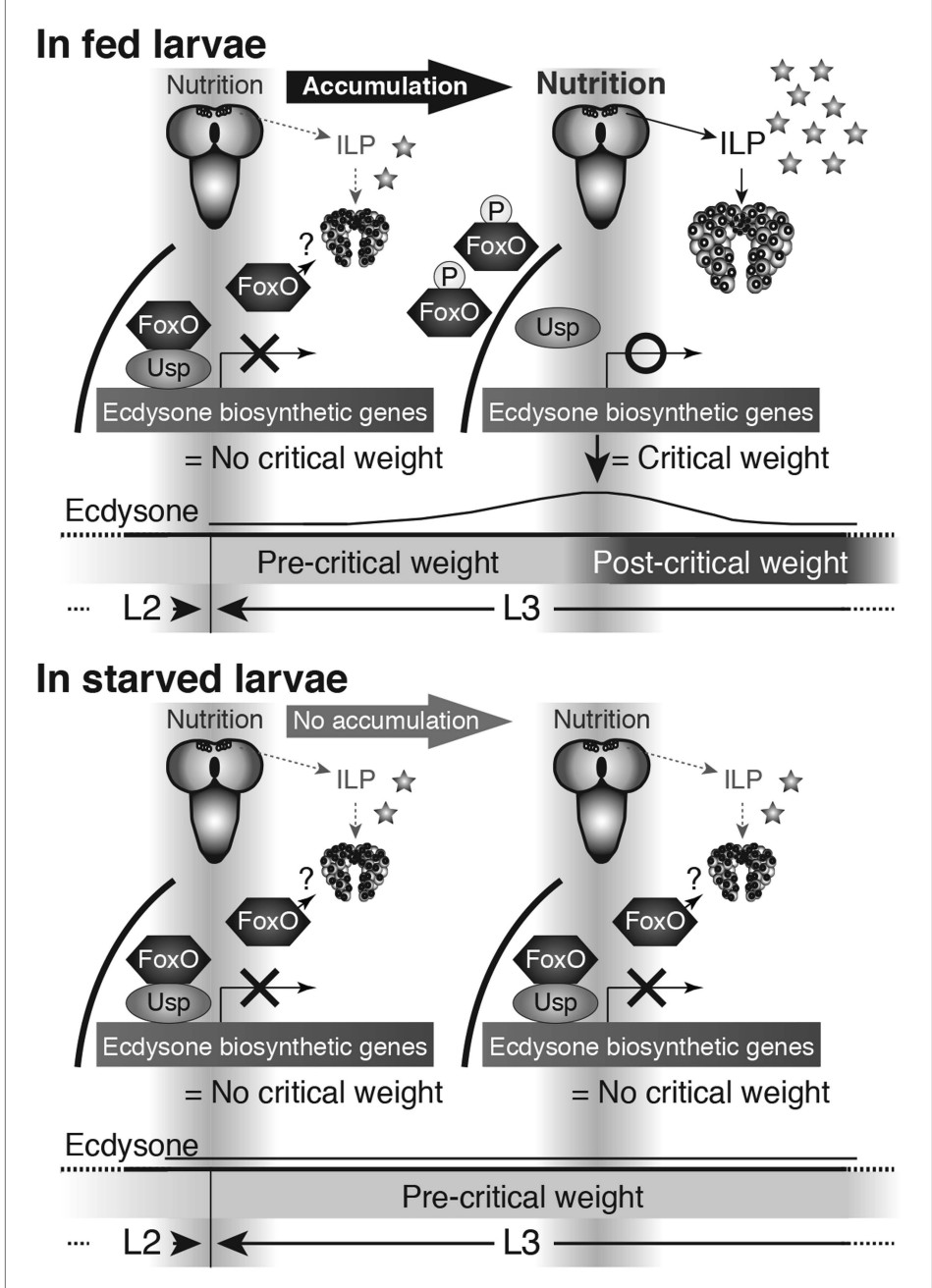

**Figure 8**. Proposed model: Nutrition regulates ecdysone biosynthesis during critical weight through FoxO/Usp. At the onset of the L3 (left), IIS/TOR signaling is reduced in the PG cells and the FoxO/Usp complex suppresses ecdysone biosynthesis either directly, as drawn, or indirectly. As the larvae feed, FoxO becomes phosphorylated and transported out of the nucleus, thereby dissociating FoxO/Usp complexes. As a result, ecdysone biosynthesis becomes derepressed (upper right). After critical weight, ecdysone reduces its own biosynthesis through a negative-feedback loop. In starved conditions, the IIS/TOR signaling activity in the PGs remains low, thereby unphosphoryl-ated FoxO remains inside of nuclei forming complexes with Usp (lower right). This inhibits ecdysone biosynthetic gene expression, thereby repressing ecdysone biosynthesis and delaying metamorphosis. FoxO on its own or with an unknown partner(s) may also regulate ecdysone biosynthesis.

*Lombardo et al., 2009*; *von Berghes et al., 2011*). These findings suggest that IIS/TOR activity regulates the production of the steroid sex hormones to regulate developmental timing in vertebrates. Furthermore, two mammalian NHRs, CAR and PXR, associate with FoxO1 to regulate the expression

of the CYP450 enzymes (*Kodama et al., 2004*). The similarity in the roles of FoxO/NHR complexes between mammals and insects provides a testable model that FoxO-NHR complexes regulate environmentally-sensitive development in a wide range of organisms.

## Materials and methods

### *Drosophila* Strains

Wild type FoxO and FoxO NK were amplified by RT-PCR using cDNA made from whole body extract of post-feeding (wandering) w[1118] larvae. After sequencing, the constructs were inserted into pUAST attB vector using *Eco*RI and *Kpn*I whose recognition sites are included on the primers, then integrated on the second chromosome by site-directed insertion using the phiC31 integrase and an attP landing site carrying recipient line, y[1] w[1118]; PBac{y[+]-attP-9A} VK00018 (Bloomington Drosophila Stock Center #9736) (*Groth et al., 2004*). w; UAS Usp 26A3 line was a gift from Dr Michael O'Connor (University of Minnesota). We used FoxO Δ94, a gift from Dr Linda Partridge (University College London), with the deficiency line, w[1118]; Df(3R)Exel8159/TM6B, Tb[1] (#7976; Bloomington) as our *FoxO null* mutant. We obtained the *PromE(800)-Gal4* (also known as *Oe-Gal4*) line from Dr Carlos Ribeiro (Champalimaud Centre for the Unknown). For Usp and FoxO knock down experiments, we used y[1] v[1]; P{y[+t7.7] v[+t1.8] = TRiP.JF02546}attP2 (#27258; Bloomington) as UAS double-stranded (ds) Usp and Vienna Drosophila RNAi Center 107786 as UAS dsFoxO.

### Larval rearing conditions, growth curves, critical weight, and pharate adult weight

Egg collections were performed on normal food plates and larvae were reared at controlled densities without additional yeast (about 200 eggs/60 mm diameter normal fly medium plate). Newly molted L3 larvae were collected every 2 hr. Collected larvae were raised in a normal cornmeal/molasses medium without additional yeast until the appropriate time point. For starvation treatments, we used 1% non-nutritional agar. To determine the duration of the L3, pupariation time was observed every 2 hr until all treated larvae pupariated or died. We defined pupariation as cessation of movement with evaginated spiracles. All treatments were performed at 25°C under constant light to avoid the effect of circadian rhythm on PTTH secretion. To analyze critical weight, we used a breakpoint analyses as previously described (*Stieper et al., 2008*; *Ghosh et al., 2013*; *Testa et al., 2013*). We constructed growth curves by weighing larvae across a range of defined ages. We then starved larvae of different age classes on non-nutritive agar media and measured the time it took for them to reach pupariation from the onset of starvation, checking for pupariation every 2 hr. We converted the time at which larvae reached critical weight to size using linear regression models from the growth curves.

For ecdysone feeding experiments, we added 0.15 mg of 20E (SciTech Chemicals, Dejvice-Hanspaulka, Czech Republic) to 1 g of normal fly medium or starvation medium (1% non-nutritive agar). After 20E was added, the media were well mixed and spun down a day before use. To measure time to pupariation from the onset of starvation, larvae were collected as above in 2 hr intervals from the molt. They were then transferred to 20E-supplemented fly medium until they reached the desired age for transfer to 20E-supplemented agar.

As a proxy for adult body size, we individually weighed pharate adults. Pharate adults, which were about 6–14 hr before eclosion, were collected from vials, carefully cleaned off using distilled water and a paint brush, and then dried for 15 min on paper towels. Once dry, pharate adults were individually weighed on an ultra-microbalance (Sartorius, SE2). We observed the presence or absence of male specific sex combs through pupal cases under a stereoscope to distinguish males from females.

### Ecdysone quantification

Concentrations of 20E were quantified using a 20-Hydroxyecdysone EIA kit (Cayman Chemicals). Carefully staged larvae were washed in distilled water twice, briefly dried on paper towels, weighed and flash-frozen on dry ice. Larvae were preserved in three-times their volume of ice cold methanol and kept at −80°C until use. Ecdysone extraction was performed as previously described (*Mirth et al., 2005*). Concentrations of 20E were quantified according to the manufacturer's instructions.

## GST-pulldown assays, western blot analysis and co-immunoprecipitation assays

Entire coding regions of *Drosophila foxo*, *ecr-A*, *ecr-B1* and *usp* cDNAs were isolated by RT-PCR using cDNA made from w[1118] wandering larvae. For *ecr-A*, *ecr-B1* and *usp* RT-PCR, forward primers were designed for gene specific sequences with a Flag-tag sequence on the 5′-end and reverse primers were designed for gene specific sequences including native stop codons. To create point mutation constructs, we designed primers containing point mutation(s) and performed standard site-directed mutagenesis methods with minor modifications. GST-tagged protein was purified by Glutathione Sepharose 4B (GE Healthcare). Flag-tagged protein was detected by the anti-Flag M2 monoclonal antibody (1:1000, Sigma). For co-immunoprecipitation assays, we used 500 µg of larval protein or cell extract, the AB11 (anti-Usp monoclonal antibody) [gifts from Drs Sho Sakurai (Kanazawa University) and Lynn M Riddiford (Janelia Farm Research Campus, HHMI)] and DDA2.7 (anti-EcR monoclonal antibody, Developmental Studies Hybridoma Bank). For western blots, the antibodies we used were: anti-Usp (1:1000, AB11), anti-EcR (1:5000, DDA2.7), anti-FoxO (1:1000) (*Puig et al., 2003*) and anti-Histone H3 (1:1000, Cell Signaling).

## Immunocytochemistry

Immunocytochemistry was performed using standard methods as described previously (*Mirth et al., 2009*). The antibodies we used were: anti-Usp (1:100, AB11), anti-FoxO (1:1000) and anti-HA (1:100, Covance). For nuclei and actin staining, we used DAPI (Invitrogen) and Phalloidin (Sigma), respectively.

## Quantitative PCR (qPCR)

Total RNA was extracted from entire larval bodies using TRIzol (Invitrogen). After DNase treatment, total RNA concentration was quantified and 1 µg total RNA was converted to cDNA using oligo dT primers and reverse transcriptase. qPCR was performed using SYBR Green PCR Master Mix (Applied Biosystems) and ABI 7900HT (Applied Biosystems). Primers are listed in *Supplementary file 2*.

## Cell culture, transfection and luciferase assays

The Dmel cell line was used for all cell culture experiments. Cells were cultured in the Express Five SMF medium (Gibco) without any serum, insulin or additives, unless mentioned. Transfection was performed using FuGENE HD Transfection Reagent (Roche), according to the manufacturer's instructions. For insulin treatment, transfected cells were re-suspended 66 hr after transfection, and split into two groups. 10 µg/ml bovine insulin (Sigma) was added into the medium of one of these groups. Cells were kept for additional 6 hr at 25°C. Luciferase assays were performed using the Luciferase Assay System (Promega), according to the manufacturer's instructions. To transfect equal amount of plasmid between all treatments, we used bacterial ampicillin resistance gene (*amp^r*). *InR*- and *4E-BP*-luciferase constructs were made according to previous study (*Puig et al., 2003*). Briefly, we designed restriction enzyme site-attached primers (*Supplementary file 3*) and amplified these promoter regions by PCR using w[1118] genomic DNA. After sequencing, we digested these fragments by *Not*I and *Bam*HI and inserted into modified pAc5-V5-His B vector (Invitrogen).

## Data availability

Data for pharate adult weight for males and females, critical weight, growth rates, qPCR and ecdysone quantifications are deposited in Dryad (doi:10.5061/dryad.75940) (*Koyama et al., 2014*). In addition, we have uploaded the scripts to generate the breakpoint plots, calculate critical size from the growth curves and to perform the permutation tests.

## Acknowledgements

We thank Drs Sho Sakurai (Kanazawa University) and Lynn M Riddiford (Janelia Farm Research Campus, HHMI) for the anti-Usp monoclonal antibody, Dr Michael B O'Connor (University of Minnesota) for UAS Usp flies, Dr Linda Partridge (University College London) for FoxO Δ94 flies and Dr Carlos Ribeiro (Champalimaud Centre for the Unknown) for *PromE(800)-Gal4* flies. We also thank Drs Michael B O'Connor, Lynn M Riddiford, Florence Janody, Élio Sucena, Patrícia Beldade, Catarina Brás-Pereira and the Mirth lab members (Instituto Gulbenkian de Ciência) for critical discussions and comments on the manuscript. This work was funded by the Fundação Calouste Gulbenkian to CKM and by the Fundação para a Ciênca e a Tecnologia to TK (SFRH/BPD/74313/2010).

# Additional information

### Funding

| Funder | Grant reference number | Author |
|---|---|---|
| Fundação para a Ciência e a Tecnologia | SFRH/BPD/74313/2010 | Takashi Koyama |
| Calouste Gulbenkian Foundation | | Christen K Mirth |

The funders had no role in study design, data collection and interpretation, or the decision to submit the work for publication.

### Author contributions

TK, Conception and design, Acquisition of data, Analysis and interpretation of data, Drafting or revising the article; MAR, Acquisition of data, Drafting or revising the article; AA, AWS, Analysis and interpretation of data, Drafting or revising the article; CKM, Conception and design, Analysis and interpretation of data, Drafting or revising the article

### Author ORCIDs

Christen K Mirth, Ⓡ http://orcid.org/0000-0002-9765-4021

# Additional files

### Supplementary files

• Supplementary file 1. Means for age and size at critical weight and time to metamorphosis from critical weight ±95% confidence intervals. Statistical comparisons for age at critical weight and size at critical weight between genotypes and treatments are shown in *Figures 3, 6* and *Figure 6—figure supplement 2*. Genotypes fed on ecdysone-supplemented medium (+20E) do not show delays in development and therefore age at critical weight cannot be determined (na). For time from critical weight to pupariation, mean times to pupariation within the same shaded box that share the same letter are statistically indistinguishable. Those that differ in letter are significantly different, as determined by Permutation tests (p < 0.034).

• Supplementary file 2. Primers used for quantitative PCR.

• Supplementary file 3. Primers used for Luciferase constructs.

### Major dataset

The following dataset was generated:

| Author(s) | Year | Dataset title | Dataset ID and/or URL | Database, license, and accessibility information |
|---|---|---|---|---|
| Koyama T, Rodrigues MA, Athanasiadis A, Shingleton AW, Mirth CM | 2014 | Data from: Nutritional Control of Body Size Plasticity through FoxO-Ultraspiracle Mediated Ecdysone Biosynthesis | doi:10.5061/dryad.75940 | Available at Dryad Digital Repository under a CC0 Public Domain Dedication. |

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
