## [Decision Letter]

Thank you for sending your work entitled “Nutritional Control of Body Size
Plasticity through FoxO-Ultraspiracle Mediated Ecdysone Biosynthesis” for
consideration at *eLife*. Your article has been favorably evaluated by
Ian Baldwin (Senior editor), a Reviewing editor, and 3 reviewers.

The Reviewing editor and the other reviewers discussed their comments before we reached
this decision, and the Reviewing editor has assembled the following comments to help you
prepare a revised submission.

Overall, all of the reviewers felt that the manuscript was interesting, largely well
done, and addressed the important issue of the achievement of “critical
weight”. The proposal that FoxO binds to Usp to form a novel complex to inhibit a
peak of ecdysone, thus regulating the critical weight transition is intriguing, and in
general the experiments demonstrating FoxO's involvement in CW are convincing.
However the reviewers agreed that additional experiments were needed to support the main
conclusions of the manuscript.

1) Show that FOXO and Usp interact in Co-IPs in fed wildtype L3 larvae. In Figure 1, it is shown that FoxO binds Usp in Co-IPs
of extracts from starved but not fed larvae. For this interaction to be relevant to the
regulation of the critical weight transition, the authors need to show that the
dFoxO-Usp interaction also occurs in fed early L3 larvae (pre-critical weight). While
the GST pulldowns with recombinant proteins in Figures 1 and 5 provide convincing evidence that FoxO and Usp have the
potential to bind in vitro, the westerns are less convincing. The westerns show a weak
Co-IPed band above the background smear that we are told corresponds to FoxO in . Given
that demonstrating this point is so essential for the main conclusion of the paper about
a molecular FoxO-Usp interaction, further controls need to be shown. One suggestion is
to show that the FoxO band (co-IPed from larval extracts with anti-Usp) present in wild
type animals is missing in FoxO null mutants.

2) Demonstrate the effects of feeding ecdysone at precritical weight times. If the model
is correct then one would expect that feeding 20E just after ecdysis to L3 will lead to
a smaller critical weight and precocious metamorphosis.

3) Concerns were raised about the specificity of the Gal4 drivers used. The P0206 driver
used as a driver to express UAS-FoxO in the PG is also expressed in the CA and in
oenocytes, which are known to regulate larval growth. The reviewers felt that it was
important to use other drivers (Aug21 for the CA and the oenocyte 'specific”
driver PromE-GAL4 from Joel Levine) as additional controls. Although the P0206-GAL4
driver has been used in similar studies of critical weight, the authors note that
phm-GAL4 driven expression of UAS-FoxO is lethal in FoxO mutants but not in control
animals. This observation may indicate that the presence of FoxO in the peripheral
tissues of control animals is modifying the phenotypic effects of PG-specific FoxO
over-expression. Although the authors suggest that animals expressing P0206-FoxO are
viable due to the moderate levels of FoxO expression, these animals could be surviving
due to the presence of FoxO in oenocytes. Therefore, the observations described in Figure 4 and Figure 6 could result from FoxO activity in either the oenocytes and not the PG or a
synergistic interaction between these two tissues. The authors may address these
problems with two simple experiments. Does oenocyte-specific expression of UAS-FoxO in a
FoxO mutant delay the onset of pupariation? And, if so, is this delay 20E dependent?
Finally, both of these Figures lack the necessary P0206-GAL4 negative control.

4) The authors' interpretations of the in vivo effects of UAS-FoxO NK (Figure 4 onwards) as a way of specifically impeding
the FoxO-Usp interaction are not necessarily correct. The experiments have been done
very thoroughly (and often sensibly using a FoxO null background) but the main result is
that the effects of UAS-FoxO NK upon critical age/weight, pupal weight, gene expression
(e74B, dib, phm) and ecdysone tend to be INTERMEDIATE between controls and UAS-FoxO
animals. This indicates that, at best, the FoxO-Usp interaction only makes a partial
contribution towards the overall FoxO activity relevant to ecdysone gene repression. At
worst, if general FoxO functions are compromised in the NK protein then the authors
would not be demonstrating any specific in vivo role for Usp-FoxO interactions, simply
that partially reducing the general activity of FoxO gives a hypomorphic phenotype.
Therefore, the authors should examine the effects of FoxONK overexpression while
simultaneously over expressing Usp. If the complex is truly important then coexpression
of foxONK together with Usp should look just like foxONK. If however, Usp coexpression
still augments the delay to critical weight, the result is more consistent for a role of
Usp in CW determination that is independent of complex formation.

Additional comments:

*Reviewer #1*:

1) The presence of ecdysone receptor (EcR) is shown to reduce the Usp-FoxO interaction
in vitro (Figure 1) but there is no follow up of
the in vivo functional significance of EcR in this context. This issue is left hanging
and so adds little to the paper. At present, the reader is left wondering whether or not
EcR directly regulates ecdysone biosynthetic gene expression in the prothoracic gland
and whether or not this regulation is antagonized by FoxO. Addressing this may add
significant extra mechanism to the model.

2) Is there any functional relevance for a well-fed larva of dFoxO/Usp repression of
ecdysone biosynthesis and its release near critical age? Loss of dFoxO activity is known
to have no obvious effect on the final size/viability of adult flies from well-fed
larvae (MA [31]). Moreover, FoxO
overexpression only delays the small L3 ecdysone peak rather than abolishing it.

3) The present manuscript will be almost impenetrable for readers outside the immediate
field and, even for insiders, it will be challenging in places. For example, many
Results sections finish with a raw result and a figure citation but they would benefit
from one sentence of interpretation/conclusion. Also, the figures and legends need
clarifying in several places such as, for Figure 4, when (and when not) FoxO null mutant backgrounds are being used in an
experiment (better indicated on the figure itself).

4) It will be confusing to readers that phm and dib expression often peak only after
critical weight is attained in controls and in some genetic manipulations (e.g. Figure 3). Moreover, the order of the phm expression
peaks for the three genotypes in Figure 3
doesn't match the order in which their critical weights are attained. How is this
observed sequence of events compatible with the model in Figure 7? This finding needs addressing as it casts doubts upon whether
increased mRNAs of ecdysone biosynthetic genes are really the trigger for critical
weight.

*Reviewer #2*:

I have a minor comment regarding Figure 2. The
authors state that critical weight in Phm>FoxO, Usp animals is significantly
increased compared with Phm>FoxO animals. This is unclear based on the bar graph,
even with the 95% confidence intervals. Providing these values in the figure legend
would provide a more convincing argument.

*Reviewer #3*:

It is unclear how the authors determine CW. The plots they provide in Figures 2 and 4 are time to pupation verses
time at which starvation is imposed. How are 'weights' determined from this
type of plot? It would seem to me that they need to show a standardization plot in which
weight gain is linear with respect to developmental time. I assume that they have done
this, but the method to determine CW should be better described in either the figure
legend or the methods.

The authors play up the fact that they have discovered a key regulator of the small
ecdysone peak that is associated with CW. In this regard, I was surprised that the
authors never mention a recent paper from the King-Jones lab (Ou Q, Magico A, King-Jones
K.PLoS Biol. 2011 Sep;9(9):e1001160.) which showed that PTTH also regulates the small
ecdysone peak through effects on the nuclear localization of the NHR HR4. Since HR4 is
proposed to negatively regulate biosynthetic enzymes prior to CW just like the FoxO/Usp
complex proposed here, it would seem to me that this warrants significant mention. Are
these two processes related or coordinated? Does HNR4 associate with FoxO? It would seem
that at some level they have to be and this should be discussed.

Lastly, In Manduca and to some extent in Drosophila, there is data suggesting that
limiting oxygen is the key to determination of critical weight. Once again there is no
discussion of this point.

---

## [Author Response]

*1) Show that FOXO and Usp interact in Co-IPs in fed wildtype L3 larvae.
In*
Figure 1*, it is shown
that FoxO binds Usp in Co-IPs of extracts from starved but not fed larvae. For this
interaction to be relevant to the regulation of the critical weight transition, the
authors need to show that the dFoxO-Usp interaction also occurs in fed early L3
larvae (pre-critical weight). While the GST pulldowns with recombinant proteins
in*
Figures 1 and 5
*provide convincing evidence that FoxO and Usp have the potential to
bind* in vitro*, the westerns are less convincing. The westerns show a
weak Co-IPed band above the background smear that we are told corresponds to FoxO in
. Given that demonstrating this point is so essential for the main conclusion of the
paper about a molecular FoxO-Usp interaction, further controls need to be shown. One
suggestion is to show that the FoxO band (co-IPed from larval extracts with anti-Usp)
present in wild type animals is missing in FoxO null mutants*.

We have performed a new Co-IP experiment examining FoxO-Usp interactions in: 1) fed
pre-critical weight wild-type larvae (0-5 h AL3E), 2) fed pre-critical weight FoxO null
larvae, 2) starved pre-critical weight wild-type larvae and 4) fed, post-critical
weight, wild-type larvae. This experiment is now in Figure 2. This experiment shows that FoxO and Usp do indeed form complexes
in fed pre-critical weight larvae as well as in protein-starved, pre-critical weight
larvae. In contrast, we do not see FoxO-Usp complexes in FoxO mutant larvae and fed,
post-critical weight larvae.

*2) Demonstrate the effects of feeding ecdysone at precritical weight times. If
the model is correct then one would expect that feeding 20E just after ecdysis to L3
will lead to a smaller critical weight and precocious metamorphosis*.

We have created a new Figure (now Figure 1)
showing that starving larvae on 20E-supplemented agar eliminated the developmental
delays seen in pre-critical weight larvae (Figure 1). We also found that larvae fed 20E supplemented fly media become smaller
in size (Figure 1—figure supplement 1).
In addition, we quantified ecdysone concentrations in starved and fed wild-type larvae
(Figure 1). We found that in starved larvae
ecdysone concentrations remain low for the first 18 hours after the moult. In fed
larvae, ecdysone concentrations peak at 10 hours after the moult. These data demonstrate
that the ecdysone peak itself is suppressed in starved pre-critical weight larvae and
that feeding ecdysone to pre-critical weight larvae is sufficient to rescue the
developmental delays induced by starvation. We hope this new data will make the
experiments that follow easier to understand.

*3) Concerns were raised about the specificity of the Gal4 drivers used. The
P0206 driver used as a driver to express UAS-FoxO in the PG is also expressed in the
CA and in oenocytes, which are known to regulate larval growth. The reviewers felt
that it was important to use other drivers (Aug21 for the CA and the oenocyte
'specific” driver PromE-GAL4 from Joel Levine) as additional controls.
Although the P0206-GAL4 driver has been used in similar studies of critical weight,
the authors note that phm-GAL4 driven expression of UAS-FoxO is lethal in FoxO
mutants but not in control animals. This observation may indicate that the presence
of FoxO in the peripheral tissues of control animals is modifying the phenotypic
effects of PG-specific FoxO over-expression. Although the authors suggest that
animals expressing P0206-FoxO are viable due to the moderate levels of FoxO
expression, these animals could be surviving due to the presence of FoxO in
oenocytes. Therefore, the observations described in*
Figure 4
*and*
Figure 6
*could result from FoxO activity in either the oenocytes and not the PG or a
synergistic interaction between these two tissues. The authors may address these
problems with two simple experiments. Does oenocyte-specific expression of UAS-FoxO
in a FoxO mutant delay the onset of pupariation? And, if so, is this delay 20E
dependent? Finally, both of these Figures lack the necessary P0206-GAL4 negative
control*.

We have used both Aug21- and PromE(800)-Gal4 to drive FoxO expression in the CA and
oenocytes of FoxO mutant animals (Figure 6—figure supplement 2). We found neither developmental time nor final
body size affected in these animals.

We’ve added the P0206>+ control to Figure 6. In Figure 7, the
P0206>+ control is in black.

*4) The authors' interpretations of the* in vivo *effects of
UAS-FoxO NK (*Figure 4
*onwards) as a way of specifically impeding the FoxO-Usp interaction are not
necessarily correct. The experiments have been done very thoroughly (and often
sensibly using a FoxO null background) but the main result is that the effects of
UAS-FoxO NK upon critical age/weight, pupal weight, gene expression (e74B, dib, phm)
and ecdysone tend to be INTERMEDIATE between controls and UAS-FoxO animals. This
indicates that, at best, the FoxO-Usp interaction only makes a partial contribution
towards the overall FoxO activity relevant to ecdysone gene repression. At worst, if
general FoxO functions are compromised in the NK protein then the authors would not
be demonstrating any specific* in vivo *role for Usp-FoxO
interactions, simply that partially reducing the general activity of FoxO gives a
hypomorphic phenotype. Therefore, the authors should examine the effects of FoxONK
overexpression while simultaneously over expressing Usp. If the complex is truly
important then coexpression of foxONK together with Usp should look just like foxONK.
If however, Usp coexpression still augments the delay to critical weight, the result
is more consistent for a role of Usp in CW determination that is independent of
complex formation*.

We agree with the reviewers that if FoxO NK is a weak hypomorph of FoxO, this would
explain the intermediate phenotype. In the first version of this manuscript, we
addressed whether FoxO NK may simply be a hypomorph of FoxO through several means.
First, we found that FoxO NK induced the expression of known FoxO target, InR and 4E-BP,
to the same levels as FoxO. Secondly, we explored whether FoxO NK overexpression reduced
the size of tissues to the same extent as FoxO. FoxO NK significantly reduced the size
of wings, and eyes, although not as much as FoxO. Importantly, FoxO NK and FoxO both
reduced the size of the prothoracic gland to the same extent.

Further, because overexpressing Usp in the prothoracic gland on its own did not
significantly change either CW or critical age, we concluded that Usp did not play an
independent role in CW. However, overexpressing both FoxO NK and Usp would provide
better evidence that Usp is not acting on its own to regulate CW. We found that
co-overexpressing Usp and FoxO NK did not affect the size and the age at critical weight
(Figure 6—figure supplement 2).
Furthermore, we did not see significant difference in body size between P0206>FoxO
NK animals and P0206>FoxO NK, Usp animals (Figure 6—figure supplement 2). Therefore, we concluded that FoxO NK bears
proper Usp-independent transcriptional activity with strongly reduced FoxO-Usp
affinity.

*Additional comments: Reviewer #1: 1) The presence of ecdysone receptor
(EcR) is shown to reduce the Usp-FoxO interaction* in vitro
*(*Figure 1*) but there is no follow up of the* in vivo
*functional significance of EcR in this context. This issue is left hanging
and so adds little to the paper. At present, the reader is left wondering whether or
not EcR directly regulates ecdysone biosynthetic gene expression in the prothoracic
gland and whether or not this regulation is antagonized by FoxO. Addressing this may
add significant extra mechanism to the model*.

In the in vitro method (GST-pulldown), we used standardized quantities of FoxO, Usp and
EcR protein that are not adjusted to the physiological levels. Our in vitro results show
that FoxO and EcR can compete for Usp binding in principle. However in vivo, our co-IP
results show that this is unlikely to be the case. We see high levels of Usp in the
co-IP overall, and in starved larvae FoxO-Usp binding does not qualitatively diminish
the amount of EcR-Usp binding (Figure 2).

*2) Is there any functional relevance for a well-fed larva of dFoxO/Usp
repression of ecdysone biosynthesis and its release near critical age? Loss of dFoxO
activity is known to have no obvious effect on the final size/viability of adult
flies from well-fed larvae (MA*
[31]*).
Moreover, FoxO overexpression only delays the small L3 ecdysone peak rather than
abolishing it*.

Our new co-IP experiment shows that FoxO and Usp make complexes in fed, pre-critical
weight larvae (Figure 2), suggesting that even
in fed larvae the complex regulates the timing of the ecdysone pulse. Starving larvae
does not abolish critical weight, but rather delays it. Thus this peak is sensitive to,
but not dependent on, nutrition. In this way, nutrition, via FoxO/Usp, can tune the
timing of the ecdysone peak at critical weight, thereby ensuring optimal growth for the
available environment.

*3) The present manuscript will be almost impenetrable for readers outside the
immediate field and, even for insiders, it will be challenging in places. For
example, many Results sections finish with a raw result and a figure citation but
they would benefit from one sentence of interpretation/conclusion. Also, the figures
and legends need clarifying in several places such as, for*
Figure 4*, when (and when
not) FoxO null mutant backgrounds are being used in an experiment (better indicated
on the figure itself)*.

Thank you for your feedback, we’ve worked hard to clarify the text and make the
results and figure legends more accessible to a broader audience.

4) It will be confusing to readers that phm and dib expression often peak only after
critical weight is attained in controls and in some genetic manipulations (e.g. Figure 3). Moreover, the order of
the phm expression peaks for the three genotypes in Figure 3 doesn't match the order in which
their critical weights are attained. How is this observed sequence of events compatible
with the model in Figure 7? This
finding needs addressing as it casts doubts upon whether increased mRNAs of ecdysone
biosynthetic genes are really the trigger for critical weight.

At 0 h AL3E, phm and dib are already high in the phm>dsFoxO, dsUsp. We presume that
this early expression is sufficient to drive ecdysone synthesis and cause premature CW
transition. Thus the peaks match the order in which CW is attained in the three
genotypes. The increases in phm and dib that occur later in this genotype is presumably
the Sgs peak. We have clarified this in the text.

*Reviewer #2: I have a minor comment regarding*
Figure 2*. The authors
state that critical weight in Phm>FoxO, Usp animals is significantly increased
compared with Phm>FoxO animals. This is unclear based on the bar graph, even
with the 95% confidence intervals. Providing these values in the figure legend would
provide a more convincing argument*.

The error bars for this graph were incorrect. We’ve changed this and also
provided a supplementary table with the values for age and size at weight for all
genotypes ([Supplementary-material SD3-data]).

*Reviewer #3: It is unclear how the authors determine CW. The plots they
provide in*
Figures 2 and 4
*are time to pupation verses time at which starvation is imposed. How are
'weights' determined from this type of plot? It would seem to me that they
need to show a standardization plot in which weight gain is linear with respect to
developmental time. I assume that they have done this, but the method to determine CW
should be better described in either the figure legend or the methods*.

You are absolutely right, we omitted both a script in the Dryad folder and a clear
explanation of how we determine size at CW. To clarify how CW is determined, we have
included a complete description of the methodology stating that we first construct
growth rate plots for each genotype (weight over age) and use linear regression to
calculate the size at CW from the age at which larvae reach CW. We have included a new
figure supplement for Figure 1 with the growth
curve for wild type (w^1118^) larvae (Figure 1—figure supplement 1), and the growth data and scripts for all
remaining genotypes in Dryad.

*The authors play up the fact that they have discovered a key regulator of the
small ecdysone peak that is associated with CW. In this regard, I was surprised that
the authors never mention a recent paper from the King-Jones lab (Ou Q, Magico A,
King-Jones K.PLoS Biol. 2011 Sep;9(9):e1001160.) which showed that PTTH also
regulates the small ecdysone peak through effects on the nuclear localization of the
NHR HR4. Since HR4 is proposed to negatively regulate biosynthetic enzymes prior to
CW just like the FoxO/Usp complex proposed here, it would seem to me that this
warrants significant mention. Are these two processes related or coordinated? Does
HNR4 associate with FoxO? It would seem that at some level they have to be and this
should be discussed*.

Yes, you are right, this was an oversight. We are focusing on how environmental cues
regulate the CW pulse, but should not neglect the lovely work from the O’Connor
and King-Jones labs on this subject.

We have added these references in the introduction, and expanded our discussion to
discuss the relative roles of insulin/TOR versus PTTH (via DHR4).

*Lastly, In Manduca and to some extent in Drosophila, there is data suggesting
that limiting oxygen is the key to determination of critical weight. Once again there
is no discussion of this point*.

We have added this to our text.